# Specific deletion of *Axin1* leads to activation of β-catenin/BMP signaling resulting in fibular hemimelia phenotype in mice

Rong Xie[1†], Dan Yi[2,3†], Daofu Zeng[2], Qiang Jie[4], Qinglin Kang[5], Zeng Zhang[5], Zhenlin Zhang[6], Guozhi Xiao[7], Lin Chen[8], Liping Tong[2*], Di Chen[2,3*]

[1]Department of Orthopedic Surgery, Rush University Medical Center, Chicago, United States; [2]Research Center for Computer-aided Drug Discovery, Shenzhen Institutes of Advanced Technology, Chinese Academy of Sciences, Shenzhen, China; [3]Faculty of Pharmaceutical Sciences, Shenzhen Institutes of Advanced Technology, Chinese Academy of Sciences, Shenzhen, China; [4]Department of Orthopedic Surgery, Honghui Hospital, Xi'an JiaoTong University, College of Medicine, Xi'an, China; [5]Department of Orthopedic Surgery, Shanghai Jiaotong University Affiliated Sixth People's Hospital, Shanghai, China; [6]Department of Osteoporosis and Bone Diseases, Shanghai Jiaotong University Affiliated Sixth People's Hospital, Shanghai, China; [7]School of Medicine, Southern University of Science and Technology, Shenzhen, China; [8]Department of Wound Repair and Rehabilitation, State Key Laboratory of Trauma, Burns and Combined Injury, Daping Hospital, Army Medical University, Chongqing, China

*For correspondence:
lp.tong@siat.ac.cn (LT);
di.chen@siat.ac.cn (DC)

†These authors contributed equally to this work

**Abstract** Axin1 is a key regulator of canonical Wnt signaling pathway. Roles of Axin1 in skeletal development and in disease occurrence have not been fully defined. Here, we report that Axin1 is essential for lower limb development. Specific deletion of *Axin1* in limb mesenchymal cells leads to fibular hemimelia (FH)-like phenotype, associated with tarsal coalition. Further studies demonstrate that FH disease is associated with additional defects in *Axin1* knockout (KO) mice, including decreased osteoclast formation and defects in angiogenesis. We then provide in vivo evidence showing that Axin1 controls limb development through both canonical β-catenin and BMP signaling pathways. We demonstrate that inhibition of β-catenin or BMP signaling could significantly reverse the FH phenotype in mice. Together, our findings reveal that integration of β-catenin and BMP signaling by Axin1 is required for lower limb development. Defect in Axin1 signaling could lead to the development of FH disease.

## Editor's evaluation

This manuscript uncovers the mesenchymal cells expressed Axin1 as a key regulator for Wnt and BMP signaling pathway which is essential for lower limb development. The data clearly demonstrates that inhibition of β-catenin and BMP signaling genetically and pharmacologically could largely reverse fibular hemimelia phenotype in mice. In general, the manuscript is clear, well-written, and concise and the study is well-structured and various techniques have been used to validate the data. It presents as a thorough study highlighting the importance of Axin1/ β-catenin/BMP signaling in FH development.

## Introduction

Fibular hemimelia (FH) is a congenital longitudinal limb deficiency characterized by complete or partial absence of the fibular bone. Unilateral fibular deficiency occurs in two-thirds of patients, with the right fibula being more often affected. FH may vary from partial absence of the fibula (10% of cases) with relatively normal-appearing limbs, to absence of the fibula with marked shortening of the femur, curved tibia, bowing of the leg, knee joint and ankle instability, and significant soft tissue deficiency. The major functional deficiency results from limb length discrepancy in patients with unilateral FH or asymmetrical dwarfism in patients with bilateral FH. The foot is generally in an equinovalgus position. As there is limited growing potential within the affected bone, the extent of the deformity tends to increase with growth.

Occasionally, FH is associated with congenital shortening of the femur. Although it was first described by Gollier in 1698, the etiology of FH remains unknown (*Stanitski and Stanitski, 2003*). The deformity of FH is probably due to disruptions during the critical period of embryonic limb development, between 4th and 7th week of gestation. Vascular dysgenesis, viral infections, trauma, and environmental influences have been suggested as possible causes. Most cases are sporadic. A family history has been reported in a small percentage of cases with an autosomal dominant pattern of inheritance and incomplete penetrance.

The evolutionary conserved canonical Wnt signaling pathway controls many biological processes during the development and maintains tissue homeostasis (*Clevers and Nusse, 2012*). A key feature of this pathway is the regulation of its downstream effector β-catenin by a cytoplasmic destruction complex. Axin1 is a central scaffold protein of the destruction complex and directly interacts with all other core components in this complex (*Clevers and Nusse, 2012*). It has been reported that Axin1 is the rate-limiting factor regulating β-catenin signaling (*Lee et al., 2003*). However, the in vivo role of Axin1 in the skeletal development and homeostasis has not been fully investigated due to early embryonic lethality (E9.5) of *Axin1* mutant mice (*Zeng et al., 1997*). Genetic evidence from both humans and mice has implicated that Wnt/β-catenin signaling plays a crucial role in controlling all major aspects of skeletal development, including craniofacial, limb, and joint formation (*Baron and Kneissel, 2013*; *Regard et al., 2012*). Bone morphogenetic protein (BMP) signaling also plays an important role in skeletogenesis during the development (*Bandyopadhyay et al., 2006*; *Yoon et al., 2005*). Thus, consistent with what is observed in many tissues and organs, Wnt and BMP signaling pathways have overlapped functions in controlling skeletal development and homeostasis. However, the key question is how the two pathways are integrated in controlling skeletal development and maintaining skeletal homeostasis.

Here, we show that loss of *Axin1* in mouse limb mesenchymal cells resulted in severe defects in lower limb development, similar to FH disease phenotype. We found that inhibition of β-catenin signaling, either by deletion of one allele of *β-catenin* gene in limb mesenchymal cells or by the treatment with a specific *β-catenin* inhibitor, was able to significantly rescue the defects in FH phenotype observed in *Axin1* knockout (KO) mice. Furthermore, inhibition of BMP signaling also significantly reversed defects in limb development and FH phenotype of *Axin1* mutant mice. Our findings indicate that Axin1/β-catenin/BMP signaling plays a key role in FH development and pathogenesis.

## Results

### Deletion of *Axin1* in *Prrx1*-expressing cells leads to FH-like phenotype

To determine the role of Axin1 in skeletal development and diseases, we generated *Axin1* conditional KO (cKO) mice by breeding the *Axin1flox/flox* mice (*Xie et al., 2011*) with *Prrx1-Cre* transgenic mice (*Logan et al., 2002*) in which the Cre expression is under the control of the *Prrx1* promoter. The *Prrx1* (paired-related homeobox gene-1) regulatory element controls Cre expression throughout the early limb bud mesenchyme and in a subset of craniofacial mesenchyme (*Logan et al., 2002*). We first determined Axin1 and β-catenin expression in 3-week-old *Axin1* cKO mice by immunohistochemistry (IHC) and found that Axin1 expression was reduced by 61% (*Figure 1A, B* *Figure 1—source data 1*; *Figure 1—source data 2*); in contrast, β-catenin expression increased 2.8-fold in joint tissue (*Figure 1C, D*, *Figure 1—source data 1*, *Figure 1—source data 3*). We examined skeletal development in E13.5 and E16.5 embryos and postnatal day 7 (P7) mice by Alizarin red/Alcian blue staining. The one notable defect is the presence of various fibular deficiencies in the *Axin1* cKO embryos and postnatal mice

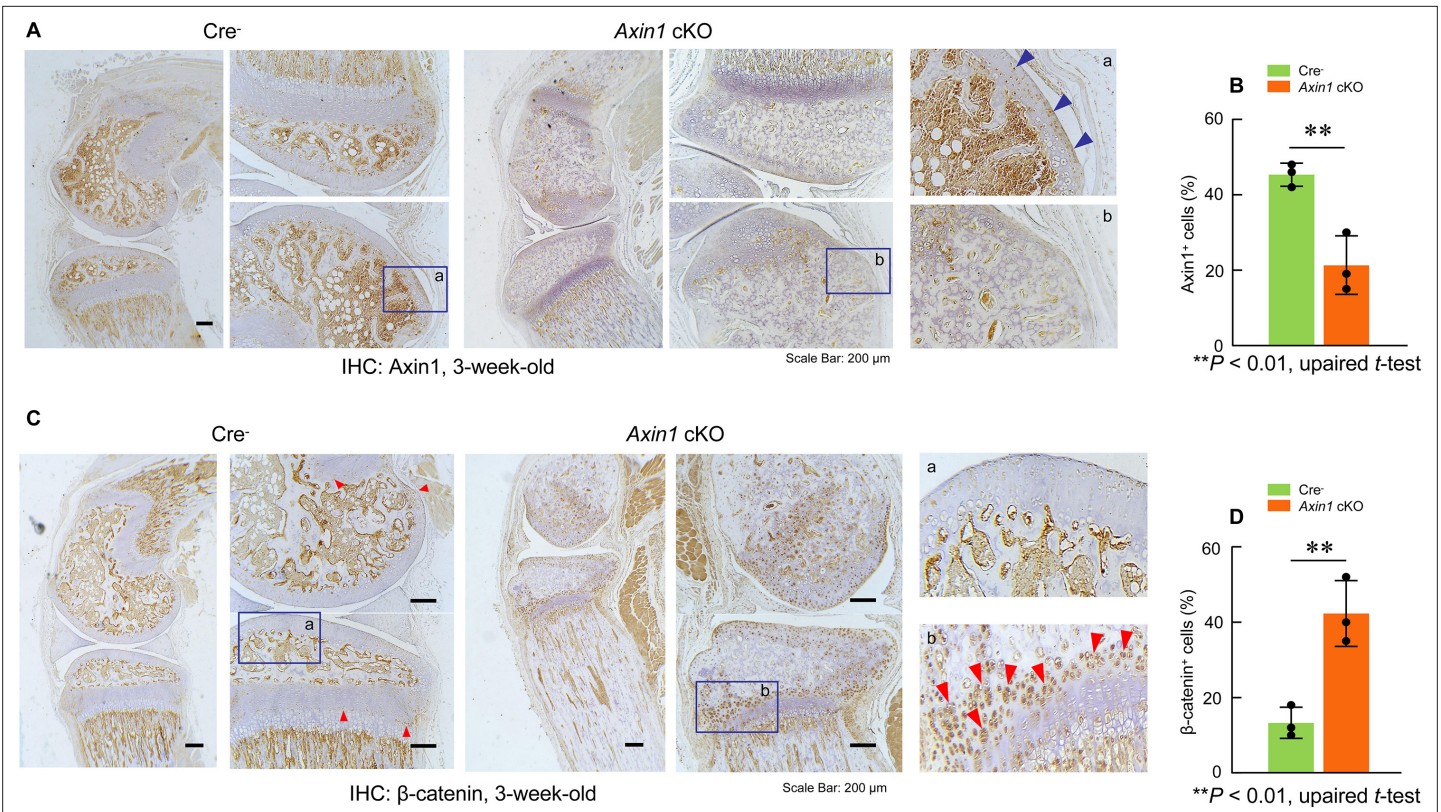

**Figure 1.** Deletion of *Axin1* in limb mesenchymal cells leads to alterations in expression of Axin1 and β-catenin proteins in long bones of *Axin1* conditional knockout (cKO) mice. We have collected long bones of 3-week-old *Axin1* cKO mice and Cre-negative littermates and examined changes in Axin1 and β-catenin protein levels by immunohistochemistry (IHC). The results demonstrated that Axin1 expression in joint tissues, such as articular cartilage (blue arrowhead), was reduced by 61% (n=3, means ± SD) (**A, B**, *Figure 1—source data 1–2*). In contrast, β-catenin expression levels in articular cartilage (red arrowheads) were increased 2.8-fold in *Axin1* cKO mice (n=3, means ± SD)(**C, D**, *Figure 1—source data 1 and 3*).

The online version of this article includes the following source data for figure 1:

**Source data 1.** Expression of Axin1 and β-catenin proteins in lower limbs of Cre-negative control and *Axin1* conditional knockout (cKO) mice.

**Source data 2.** Original numbers used for quantification of percentage of Axin1-positive cells in lower limbs in Cre-negative control and *Axin1* conditional knockout (cKO) mice.

**Source data 3.** Original numbers used for quantification of percentage of β-catenin-positive cells in lower limbs in Cre-negative control and *Axin1* conditional knockout (cKO) mice.

(*Figure 2A*, *Figure 2—source data 1*). The fibulae of *Axin1* cKO mice did not mineralize even at P7. Histological analysis using limb tissues dissected from *Axin1* cKO mice showed partially developed fibular tissues with poorly developed growth plate (*Figure 2B*, *Figure 2—source data 1*). The number of chondrocytes was significantly reduced and the structure of growth plate was disorganized in *Axin1* cKO mice (*Figure 2B*, *Figure 2—source data 1*). In addition to fibular defects, the lengths of femorae and tibiae were significantly reduced in *Axin1* cKO mice (*Figure 2C–E*, *Figure 2—source data 1–3*). High bone mass phenotype was observed in all long bones, including femorae, tibiae, and fibulae in *Axin1* cKO mice (*Figure 2B, C*, *Figure 2—source data 1*). To date, we have analyzed 52 *Axin1* cKO mice and all of them have a fibular deficiency phenotype. Radiographic analysis of 4- and 8-week-old mice showed that some of *Axin1* cKO mice were completely or almost completely absence of fibulae (>50% loss of fibulae, 27/52) where only a distal, vestigial fragment was present. The other *Axin1* cKO mice had partial absence of the fibulae (30–50% loss of fibulae, 23/52) in which the proximal portions of the fibulae were absent while distal portions were present but could not support the ankle (*Figure 2F, G*, *Figure 2—source data 1*). The mild fibular defects were observed in few of *Axin1* cKO mice (2/52), in which the fibulae were absent less than 30% of their normal length (*Figure 2H*, *Figure 2—source data 1*). These results demonstrate that Axin1 plays an essential role in fibular development. In addition, all femorae of *Axin1* cKO mice were shorter and wider than those of their

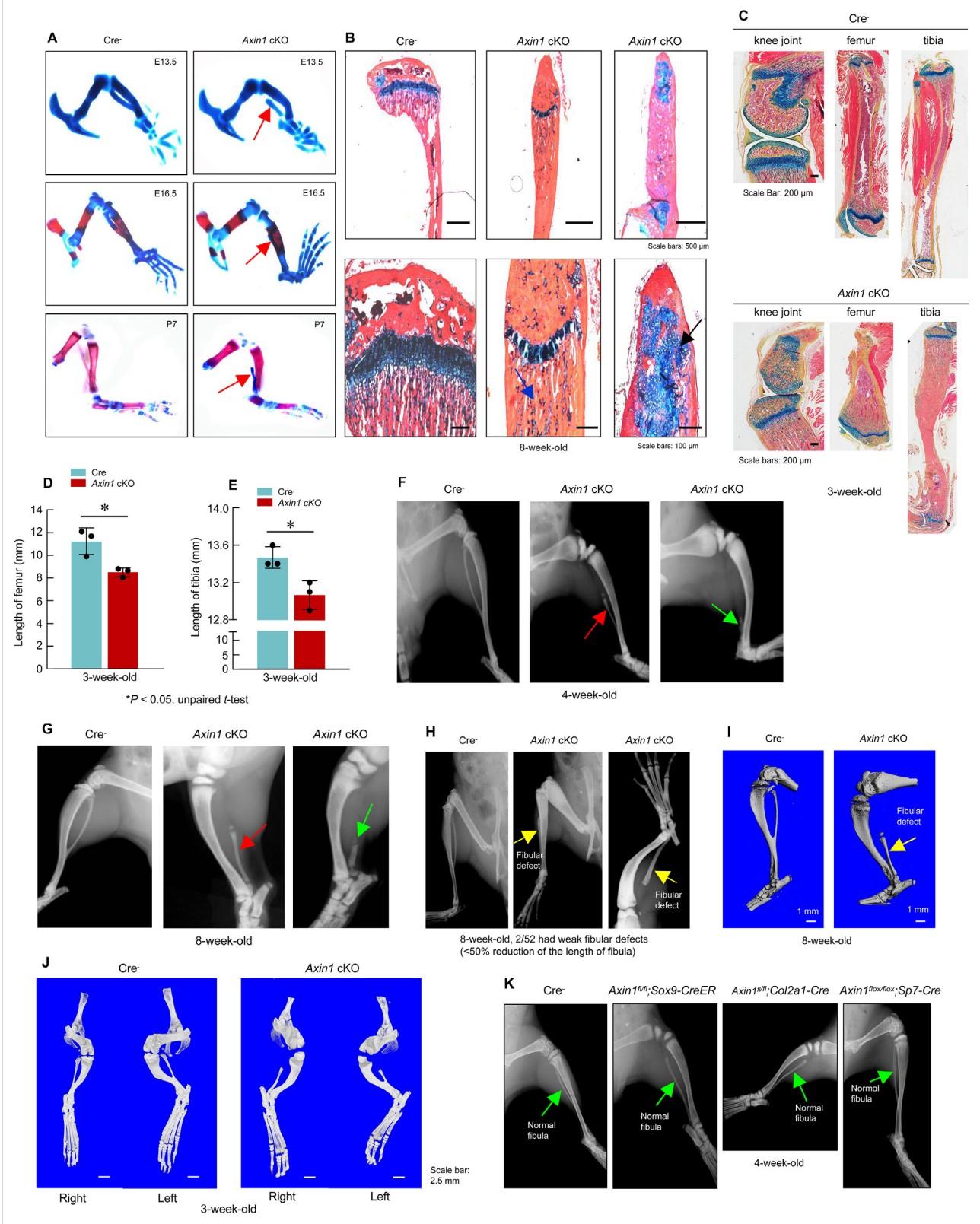

**Figure 2.** Deletion of *Axin1* in limb mesenchymal cells leads to defects resembling to fibular hemimelia (FH) disease. (**A**) *Axin1* conditional knockout (cKO) embryos and postnatal mice showed partial development of fibula, which did not mineralize even at day 7 of postnatal stage (red arrows, ***Figure 2—source data 1***). (**B**) Histological analysis showed that disorganized fibular structure (black arrow) and abnormal cartilage development were observed in 8-week-old *Axin1* cKO mice (***Figure 2—source data 1***). (**C–E**) In addition, the lengths of tibiae and femorae were significantly decreased in 3-week-

*Figure 2 continued on next page*

*Figure 2 continued*

old *Axin1* cKO mice (n=3, means ± SD) (***Figure 2—source data 1–3***). (**F, G**) We have generated and analyzed 52 *Axin1* cKO mice (***Figure 2—source data 1***). Defects in fibular development were observed in all 52 *Axin1* cKO mice that we have analyzed. Radiographic analysis showed that the fibula in some of *Axin1* cKO mice was almost completely absent (>50% loss, 27/52) where only a distal, vestigial fragment was present (green arrow, right panel). The other *Axin1* cKO mice had partial absence of the fibula (30–50% loss, 23/52) (red arrow, middle panel) in which the proximal portion of the fibula was absent while the distal portion was present in 4-week-old *Axin1* cKO mice. (**H**) Radiographic and microcomputed tomography (μCT) analyses showed that fibulae were developed over 50% of their length in few *Axin1* cKO mice (2/52) (***Figure 2—source data 1***). (**I**) Results of μCT analysis also showed fibular defects in 8-week-old *Axin1* cKO mice (***Figure 2—source data 1***). (**J**) μCT analysis also showed that fibular defects were observed in hindlimbs at both right and left sides (***Figure 2—source data 1***). (**K**) To determine the role of Axin1 in other cell populations, we generated *Axin1* cKO mice in other cell types, such as *Sox9*-expressing cells (*Axin1^flox/flox^;Sox9-CreER*), *Col2a1*-expressing cells (*Axin1^flox/flox^;Col2a1-Cre*), and *Sp7*-expressing cells (*Axin1^flox/flox^;Sp7-Cre*). X-ray radiographic analysis showed that deletion of *Axin1* in *Sox9*-, *Col2a1*-, and *Sp7*-expressing cells did not affect lower limb development (***Figure 2—source data 1***).

The online version of this article includes the following source data for figure 2:

**Source data 1.** Histology, X-ray, and microcomputed tomography (μCT) analysis in Cre-negative control and *Axin1* conditional knockout (cKO) mice.

**Source data 2.** Original lengths for quantification of femorae of Cre-negative control and *Axin1* conditional knockout (cKO) mice.

**Source data 3.** Original lengths for quantification of tibiae of Cre-negative control and *Axin1* conditional knockout (cKO) mice.

Cre⁻ littermates (***Figure 2H, I***, ***Figure 2—source data 1***). Different from human FH disease, *Axin1* cKO mice always displayed fibular defects in both sides of lower limb (***Figure 2J***, ***Figure 2—source data 1***). It is also interesting to note that the bowed tibiae were observed in *Axin1* cKO mice (***Figure 2G–J***, ***Figure 2—source data 1***). These skeletal defects observed in *Axin1* cKO mice have been reported to be the key features of FH disease in humans (***Achterman and Kalamchi, 1979***). In contrast to *Axin1* cKO mice, deletion of *Axin1* in *Sox9*-, *Col2a1*-, and *Sp7*-expressing cells does not affect fibular development (***Figure 2K***, ***Figure 2—source data 1***), suggesting that FH disease is caused by defects in the specific cell population, limb mesenchymal cells.

## Functional defects in lower limb in *Axin1* cKO mice

In addition to fibular defects, *Axin1* cKO mice also had femoral defect phenotype. The dysplasia of acetabulum and defects in femoral head development were found by microcomputed tomography (μCT) analysis in *Axin1* cKO mice (***Figure 3A***, ***Figure 3—source data 1***). We performed X-ray and μCT analyses on knee joint and found hypoplasia of knee joint in 3-week-old *Axin1* cKO mice (***Figure 3B***, ***Figure 3—source data 1***). The tarsal coalition phenotype was also found in *Axin1* cKO mice and distal tarsals 2, 3, and 4 were fused together (***Figure 3C, D***, ***Figure 3—source data 1***). It is known that Wnt/β-catenin signaling regulates osteoclast formation. We then performed TRAP staining and examined changes in TRAP-positive osteoclast numbers in *Axin1* cKO mice and found a significant decrease in osteoclast formation in *Axin1* cKO mice (***Figure 3E, F***, ***Figure 3—source data 1–2***). Inhibition of osteoclast formation may contribute to the high bone mass phenotype observed in *Axin1* cKO mice. We also performed immunofluorescent (IF) staining of VEGF and found that expression of VEGF was significantly reduced in joint tissues of 3-week-old *Axin1* cKO mice (***Figure 3G, H***, ***Figure 3—source data 1***, ***Figure 3—source data 3***). Reduced VEGF expression may be related to the hypoplasia phenotype of lower limb in *Axin1* cKO mice.

## Inhibition of β-catenin signaling reverses FH defects in *Axin1* cKO mice

Since Axin1 is a well-known negative regulator of canonical Wnt pathway, deletion of *Axin1* elevated β-catenin protein levels. We think if defects in fibular development in *Axin1* cKO mice are due to elevated levels of β-catenin, reducing the expression levels of β-catenin may fully or partially correct defects observed in *Axin1* cKO mice. To test this hypothesis, we examined genetic interaction between Axin1 and β-catenin during skeletal development in double mutant mice (*Axin1^flox/flox^; β-catenin^flox/wt^; Prrx1-Cre*). We found that deletion of one allele of the *β-catenin* gene under *Axin1* cKO background significantly reversed defects in fibular development (***Figure 4A, B***, ***Figure 4—source data 1***) and caused reduction of BV from 92% to 71% (***Figure 4C***, ***Figure 4—source data 2***). In addition, we also used a specific β-catenin inhibitor iCRT14 to determine if blocking β-catenin signaling could reverse defects in the fibular development observed in *Axin1* cKO mice. It has been shown that iCRT14 specifically inhibits β-catenin-induced transcription by disrupting the interaction between β-catenin and TCF4 (***Gonsalves et al., 2011***). The iCRT14 (2.5 mg/kg) was injected into *Axin1* cKO mice (pregnant

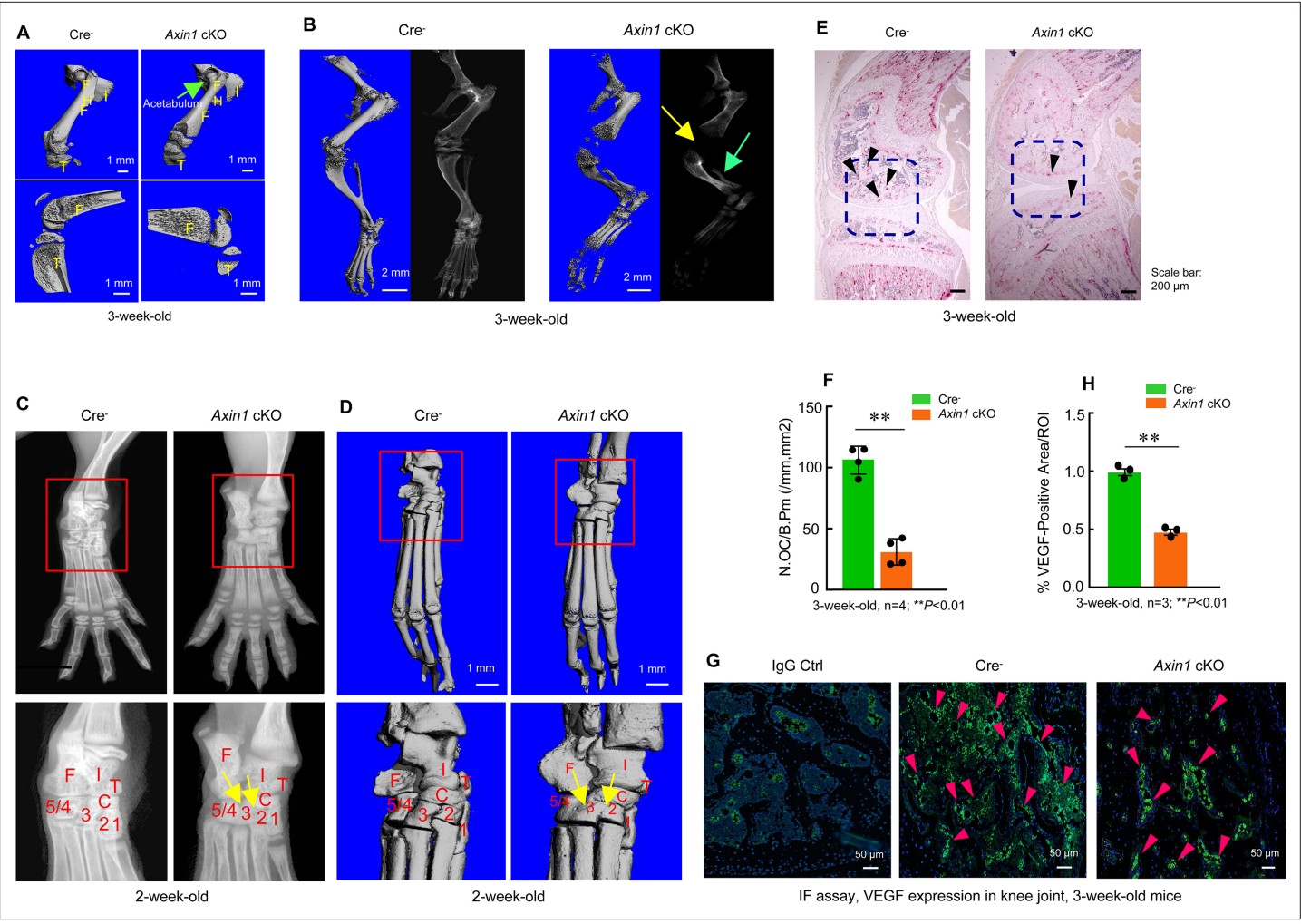

**Figure 3.** *Axin1* conditional knockout (cKO) mice had multiple defects in lower limb. (**A**) Microcomputed tomography (µCT) analysis showed that *Axin1* KO mice (3-week-old) had defects in femerae including increased width of femorae and the dysplasia of acetabulum was found in 3-week-old *Axin1* cKO mice (*Figure 3—source data 1*). (**B**) µCT and X-ray analyses showed that defects in fibular and joint development in 3-week-old *Axin1* cKO mice (*Figure 3—source data 1*). (**C, D**) X-ray and µCT analyses showed that the tarsal elements 2, 3, and 4 of ankle joint were fused in 2-week-old *Axin1* cKO mice (*Figure 3—source data 1*). (**E, F**) We performed TRAP staining and found that reduction in osteoclast numbers in 3-week-old *Axin1* cKO mice (*Figure 3—source data 1–2*). (**G, H**) Expression of VEGF was analyzed by immunofluorescent (IF) staining and significant decrease in VEGF expression was found in 3-week-old *Axin1* cKO mice (*Figure 3—source data 1 and 3*).

The online version of this article includes the following source data for figure 3:

**Source data 1.** Microcomputed tomography (µCT), X-ray, TRAP staining, and immunofluorescence (IF) analysis of Cre-negative control and *Axin1* conditional knockout (cKO) mice.

**Source data 2.** Original numbers used for quantification of osteoclast numbers in Cre-negative control and *Axin1* conditional knockout (cKO) mice.

**Source data 3.** Original numbers used for quantification of percentage of VEGF-positive cells in Cre-negative control and *Axin1* conditional knockout (cKO) mice.

mothers at E9.5 stage, i.p. injection). The embryos were collected at E18.5. Histological analysis showed that the fibular defect phenotype in *Axin1* cKO embryos were rescued by the treatment with iCRT14 (*Figure 4D*, *Figure 4—source data 1*). The rescuing of fibular defects with iCRT14 treatment was also confirmed by µCT analysis in 4-week-old *Axin1* cKO mice receiving iCRT14 treatment (*Figure 4E*, *Figure 4—source data 1*). It is interesting to note that inhibition of Wnt secretion with LGK974 did not rescue the fibular defects observed in *Axin1* cKO mice (*Figure 4F*, *Figure 4—source data 1*). LGK974 is a specific small molecule compound of porcupine inhibitor, which inhibits Wnt secretion in vitro and in vivo (*Liu et al., 2013*). Together, these results demonstrate that Axin1 controls the fibular development through the canonical β-catenin signaling pathway.

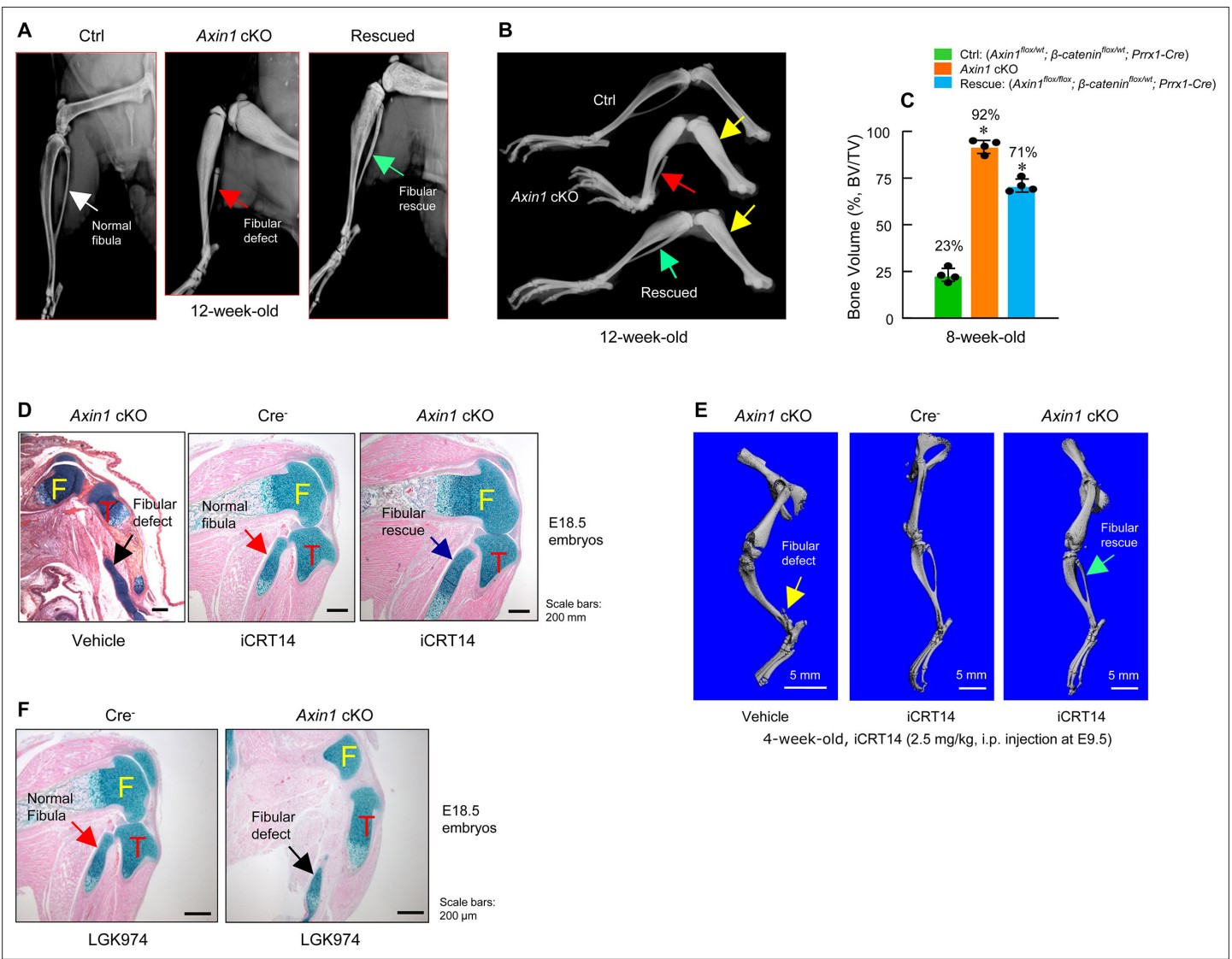

**Figure 4.** Inhibition of β-catenin signaling reverses defects in skeletal development observed in *Axin1* conditional knockout (cKO) mice. (**A, B**) X-ray radiographic analysis showed that defects in fibular development observed in *Axin1* cKO mice (red arrows) were significantly reversed (green arrows) by deletion of one allele of *β-catenin* gene (rescued) (12-week-old mice) (*Figure 4—source data 1*). Yellow arrows showed bone mass was increased in *Axin1* cKO mice and was partially reversed in rescued mice. Radiographic analysis showed that deletion of one allele of *β-catenin* gene significantly reversed defects in fibular development in 12-week-old *Axin1* cKO mice. (**C**) Microcomputed tomography (μCT) analysis of hindlimbs of 8-week-old mice, including Ctrl: (*Axin1^{flox/wt};β-catenin^{flox/wt};Prrx1-Cre*), *Axin1* cKO, and rescued: (*Axin1^{flox/flox};β-catenin^{flox/wt};Prrx1-Cre*) mice, showed that bone volume was reduced from 92% in *Axin1* cKO mice to 71% in rescued mice (n=4) (*Figure 4—source data 2*). (**D**) Results of histological analysis of E18.5 embryos showed that treatment with β-catenin inhibitor, iCRT14 (2.5 mg/kg, i.p. injection to the pregnant mothers at E9.5 stage), almost completely reversed defects in fibular development in *Axin1* cKO embryos (*Figure 4—source data 1*). (**E**) μCT analysis confirmed that the treatment with iCRT14 reversed defects in lower limb development in *Axin1* cKO mice, such as lack of fibula and bowed tibia (*Figure 4—source data 1*). (**F**) In contrast, the treatment with LGK974 (inhibitor of Wnt secretion) failed to reverse fibular hemimelia (FH) phenotype observed in *Axin1* cKO mice. Data presented in (C) were analyzed by one-way ANOVA followed by the Tukey's post hoc test (n=4, means ± SD, *p<0.05) (*Figure 4—source data 1*).

The online version of this article includes the following source data for figure 4:

**Source data 1.** X-ray, histology, and microcomputed tomography (μCT) analysis of *Axin1* conditional knockout (cKO) mice with or without iCRT14 treatment.

**Source data 2.** Original data used to quantify the percentage of bone volume in *Axin1* conditional knockout (cKO) mice with or without iCRT14 treatment.

## Inhibition of BMP signaling reverses fibular defects in *Axin1* cKO mice

In previous studies, we found that *Bmp2* and *Bmp4* expression was upregulated in *Axin2* KO mice (*Yu et al., 2005*; *Yan et al., 2009*). To determine if BMP signaling is upregulated in *Axin1* cKO mice, we extracted total RNA from hindlimbs derived from E12.5 Cre⁻ and *Axin1* cKO embryos. We found that expression of *Bmp2*, *Bmp4*, *Gremlin1*, and *Msx2* was significantly upregulated in limb tissues of *Axin1* cKO embryos (*Figure 5A–E*, *Figure 5—source data 1–5*). To determine if inhibition of BMP signaling will reverse fibular defects observed in *Axin1* cKO mice, we injected *Axin1* cKO mice with BMP signaling inhibitor dorsomorphin (2.5 mg/kg, i.p. injection) to the pregnant female mice at E9.5 stage. Dorsomorphin has been shown to inhibit BMPR-IA (ALK3), BMPR-IB (ALK6), and ALK2 activity (*Yu et al., 2008*). Analysis of histological sections of hindlimbs of E18.5 embryos showed that fibular defects in *Axin1* cKO embryos were significantly rescued by the treatment with dorsomorphin (*Figure 5F*, *Figure 5—source data 6*). The result of µCT analysis of 6-week-old mice confirmed that the fibular defect phenotype observed in *Axin1* cKO mice was significantly reversed by the treatment with dorsomorphin (*Figure 5G*, *Figure 5—source data 6*). In contrast, single dose of BMP inhibitor (2.5 mg/kg at E9.5 stage) was not able to reverse high bone mass phenotype caused by continuing upregulation of BMP signaling in *Axin1* cKO mice. Also, dorsomorphin did not affect fibular development in the Cre⁻ control mice (*Figure 5F, G*, *Figure 5—source data 6*). We also determined the stage-specific effect of dorsomorphin on reversing fibular defects in 3-week-old *Axin1* cKO mice and found that dorsomorphin lost its protective effect if injected after embryonic E12.5 stage (*Figure 5H*, *Figure 5—source data 6*). Administration of dorsomorphin at E12.5 stage significantly reversed fibular defects as well as rescuing knee join dysplasia phenotype in 3-week-old *Axin1* cKO mice (*Figure 5I*, *Figure 5—source data 6*). In contrast, same concentration of TGF-β inhibitor, small chemical compound SB-505124, had no effect on reversing fibular defect in *Axin1* cKO mice (*Figure 5J*, *Figure 5—source data 6*). These results demonstrate that BMP signaling upregulation also contributes to the FH defects observed in *Axin1* cKO mice.

## Axin1 inhibits BMP signaling through promoting pSmad5 degradation

Next, we sought to explore if Axin1 directly regulates BMP signaling although it is known that BMP signaling is downstream of β-catenin signaling in bone cells. Since Axin1 serves as scaffold protein, we examined whether there are interactions between Axin1 and Smad proteins. The results of co-immunoprecipitation (co-IP) assays revealed that endogenous Axin1 indeed interacted with Smad5 in C3H10T1/2 cells (*Figure 6A*, *Figure 6—source data 1*). Then, we investigated whether Axin1 regulates the stability of pSmad5. In pulse-chase experiments, the limb cells from E12.5 Cre⁻ or *Axin1* cKO embryos were treated with BMP2 for 0.5 hr, followed by incubation without BMP to track the levels of phosphorylated Smad5 (pSmad5) (*Figure 6B, C*, *Figure 6—source data 2–3*). The pSmad5 levels decreased gradually after removal of BMP2 in Cre⁻ cells, but duration of pSmad5 was much longer in *Axin1* mutant cells (*Figure 6B, C*, *Figure 6—source data 2–3*). The results indicate that Axin1 could also inhibit BMP signaling through promoting pSmad5 degradation. We next determined if the increased duration of pSmad5 by *Axin1* deletion is independent of β-catenin. We performed BMP2-induced pulse-chase experiments in the presence of iCRT14. The treatment with iCRT14 did not affect the duration of pSmad5 in Cre⁻ cells as well as in *Axin1*-deficient cells (*Figure 6D*, *Figure 6—source data 4*). In contrast, LGK974 did block the Wnt3a-induced prolonged duration of pSmad5 (*Figure 6D*, *Figure 6—source data 4*). These results suggest that Axin1 regulated pSmad5 levels are independent from β-catenin.

To determine whether Wnt3a-induced pSmad5 increase occurs through inhibition of its degradation, serum-starved C3H10T1/2 cells were stimulated with BMP2 for 0.5 hr. After BMP2 was removed, the cells were treated with or without Wnt3a for 3 hr. Indeed, Wnt3a significantly prolonged the duration of the pSmad5 at the similar extent as the treatment with proteasome inhibitor MG132 (*Figure 6E*, *Figure 6—source data 5–6*), and iCRT14 did not block the Wnt3a-induced prolonged duration of pSmad5 in C3H10T1/2 cells (*Figure 6E*, *Figure 6—source data 5–6*). It is worth noting that total Smad5 levels did not change under these experimental conditions, suggesting that only a small fraction of Smad5 is phosphorylated in response to the treatment of BMP2 at the physiological level. These results are in agreement with previous observations (*Fuentealba et al., 2007*; *Alarcón et al., 2009*). Taken together, these data suggest that Axin1 could regulate BMP signaling through direct and indirect mechanisms.

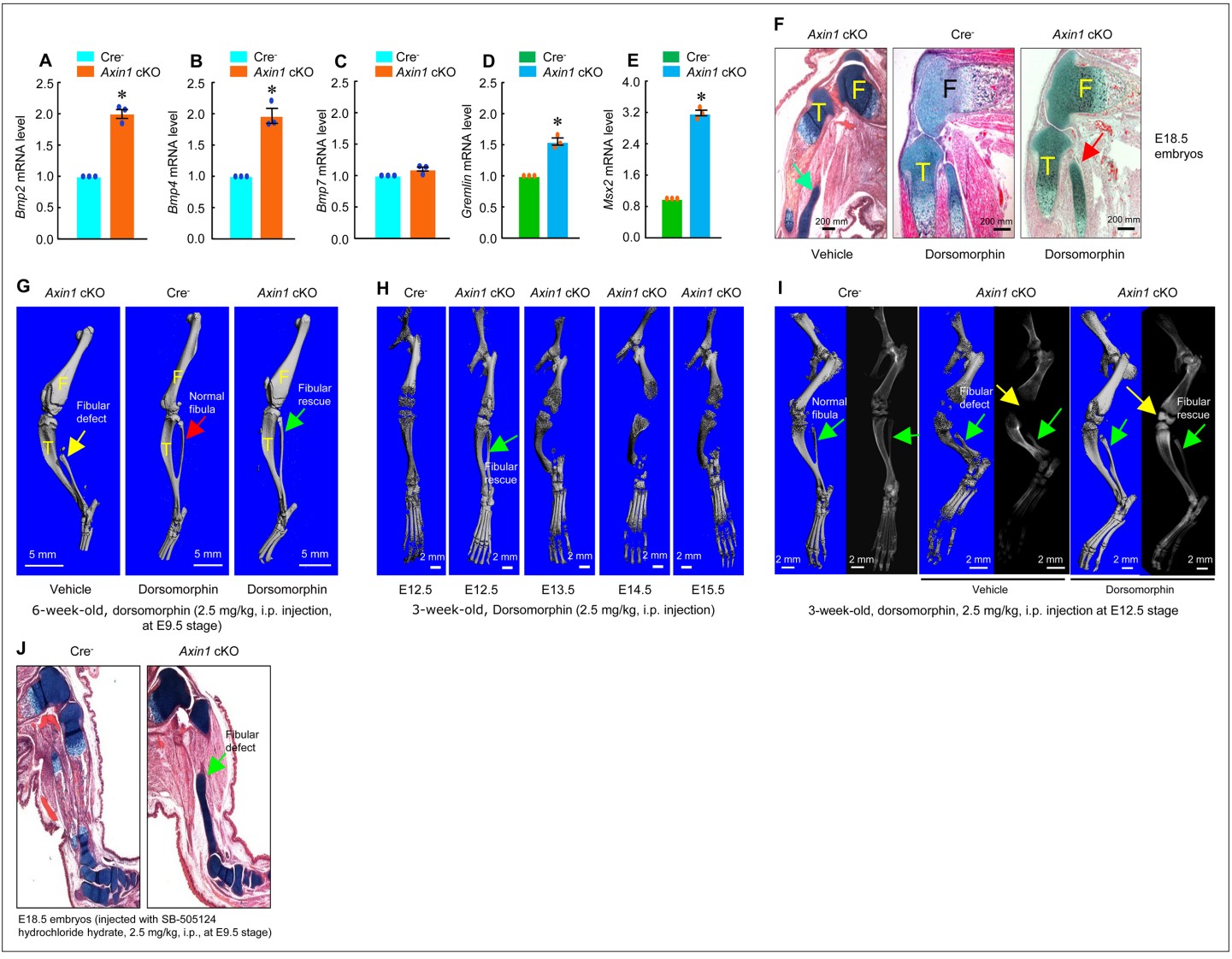

**Figure 5.** Inhibition of bone morphogenetic protein (BMP) signaling significantly reversed skeletal phenotype observed in *Axin1* conditional knockout (cKO) mice. (**A–E**) BMP signaling was upregulated in limb bud cells derived from E12.5 *Axin1* cKO embryos. Real-time PCR analysis showed that expression of *Bmp2, Bmp4, Bmp7, Gremlin,* and *Msx2* was significantly upregulated in limb bud cells derived from E12.5 *Axin1* cKO embryos (*$p<0.05$, unpaired Student's *t*-test, n=3, means ± SD) (*Figure 5—source data 1–5*). (**F**) Inhibition of BMP signaling by the treatment with BMP inhibitor, dorsomorphin (2.5 mg/kg, i.p. injection to E9.5 mothers), reversed fibular defects of *Axin1* cKO embryos (E18.5) (*Figure 5—source data 6*). Green arrow (left panel): the defect in the fibular development. (**G**) Microcomputed tomography (μCT) analysis showed that the treatment with dorsomorphin (2.5 mg/kg, i.p. injection to E9.5 mothers) completely reversed defects in fibular development in 6-week-old *Axin1* cKO mice (*Figure 6—source data 6*). (**H**) We also examined stage-specific effect of dorsomorphin treatment and found that dorsomorphin could effectively reverse defects in fibular development up to E12.5. No rescuing effect could be observed when dorsomorphin was administered at later stages, such as E13.5, E14.5, and E15.5 (*Figure 5—source data 6*). (**I**) Dorsomorphin could significantly reverse defects in fibular development when injected at E12.5 stage (*Figure 5—source data 6*). (**J**) In contrast, injection of same amount of TGF-β inhibitor SB-505124 had no significant effect on fibular development (*Figure 5—source data 6*).

The online version of this article includes the following source data for figure 5:

**Source data 1.** Original numbers used for quantification of RNA relative *Bmp2* expression in Cre-negative control and *Axin1* conditional knockout (cKO) mice.

**Source data 2.** Original numbers used for quantification of RNA relative *Bmp4* expression in Cre-negative control and *Axin1* conditional knockout (cKO) mice.

**Source data 3.** Original numbers used for quantification of RNA relative *Bmp7* expression in Cre-negative control and *Axin1* conditional knockout (cKO) mice.

*Figure 5 continued on next page*

*Figure 5 continued*

**Source data 4.** Original numbers used for quantification of RNA relative expression of *Gremlin* in Cre-negative control and *Axin1* conditional knockout (cKO) mice.

**Source data 5.** Original numbers used for *Figure 5—source data 5* quantification of RNA relative expression of *Msx2* in Cre-negative control and *Axin1* conditional knockout (cKO) mice.

**Source data 6.** Histology, microcomputed tomography (μCT), and X-ray analysis of *Axin1* conditional knockout (cKO) mice treated with or without dorsomorphin or SB-505124.

## Discussion

FH is the most common deficiency of long bone and the pathological mechanisms of FH are currently unknown. Our present study clearly demonstrated that Axin1 plays a key role in lower limb development and FH pathogenesis through regulation of both β-catenin and BMP signaling. FH is a birth defect where part of or entire fibulae are missing, as well as associated with limb length discrepancy and foot and knee deformities. FH is a rare genetic disorder, occurring in about 1 in 40,000 live births. In the present studies, we found that the FH phenotype was observed in *Axin1* cKO (*Axin1^{flox/flox}*;*Prrx1-Cre*) mice, but not in *Axin1* cKO mice targeting in other cell populations, such as *Sox9*-expressing cells (*Axin1^{flox/flox}*;*Sox9-CreER*), *Col2a1*-expressing cells (*Axin1^{flox/flox}*;*Col2a1-Cre*), and *Sp7*-expressing cells (*Axin1^{flox/flox}*;*Sp7-Cre*), suggesting that specific deficiency of *Axin1* in *Prrx1*-expressing cell population is responsible for the FH development. In addition, we also found that β-catenin or BMP inhibitor can only rescue the FH phenotype during E9.5-E12.5 developmental stages. These findings suggest that FH disease may be caused by specific upregulation of β-catenin or BMP signaling in limb mesenchymal cells during early stage of skeletal development.

The major defect observed in *Axin1* cKO mice is the presence of various fibular deficiencies. Some of *Axin1* cKO mice show complete or almost complete absence of fibula. The other *Axin1* cKO mice had partial absence of fibulae. As we described above, the partial or total absence of fibulae are the key feature of FH. The *Axin1* cKO mice also exhibit tarsal coalition.

As Axin1 is a key negative regulator of canonical Wnt pathway and β-catenin is a major downstream mediator of Wnt signaling, we determined genetic interactions between Axin1 and β-catenin during lower limb development. Indeed, the defects in *Axin1* cKO mice are significantly rescued by the deletion of one allele of the *β-catenin* gene, *Ctnnb1*. Furthermore, we found that β-catenin inhibitor iCRT14 significantly reversed the FH phenotype in *Axin1* cKO mice. These results strongly suggest that Axin1 controls lower limb development through canonical β-catenin pathway.

BMP signaling may be a downstream signaling pathway of Wnt/Axin1 signaling during limb development. In this study we found upregulation of BMP signaling in *Axin1* cKO mice. More importantly, we found that inhibition of BMP signaling efficiently rescued the FH phenotype observed in *Axin1* cKO mice. These results indicate that BMP signaling pathway is another key downstream effector of Wnt/Axin1 signaling during limb development. Therefore, it is clear that Axin1 controls lower limb development through both canonical β-catenin and BMP signaling pathways. It is known that the Wnt and BMP signaling pathways coordinately govern many developmental processes. However, the mechanisms by which these two pathways are integrated each other during skeletal development remain elusive. We found that pSmad5 stability is significantly prolonged in *Axin1*-deficient cells. And we demonstrated that increased pSmad5 stability in *Axin1*-deficient cells is independent from β-catenin activity. Next, we found that Wnt indeed regulates Smad5 phosphorylation in C3H10T1/2 cells, in consistent with previous reports (*Fuentealba et al., 2007*). Importantly, we demonstrate that this regulation is independent from β-catenin signaling. In addition, we confirmed that Axin1 directly interacts with Smad5 in C3H10T1/2 cells. Taken together, we conclude that Axin1 is not only a key negative regulator of β-catenin, but also simultaneously a critical negative regulator of Smad5.

Here, we propose a mechanism by which integration between Wnt and BMP signaling pathways by Axin1 controls limb development and homeostasis. In the absence of Wnt stimulation, β-catenin and Smad5 are degraded by the same destruction complex through interaction with Axin1. In the presence of Wnt ligands or loss of Axin1, both β-catenin and Smad5 degradation is inhibited, resulting in activation of β-catenin and Smad5 signaling (*Figure 6F*). It is well established that LRP6 recruits the active destruction complex to form the signaling complex with Wnt stimulation, which results in inactivation of the destruction complex, leading to β-catenin stabilization (*Li et al., 2012*; *Kim et al.,*

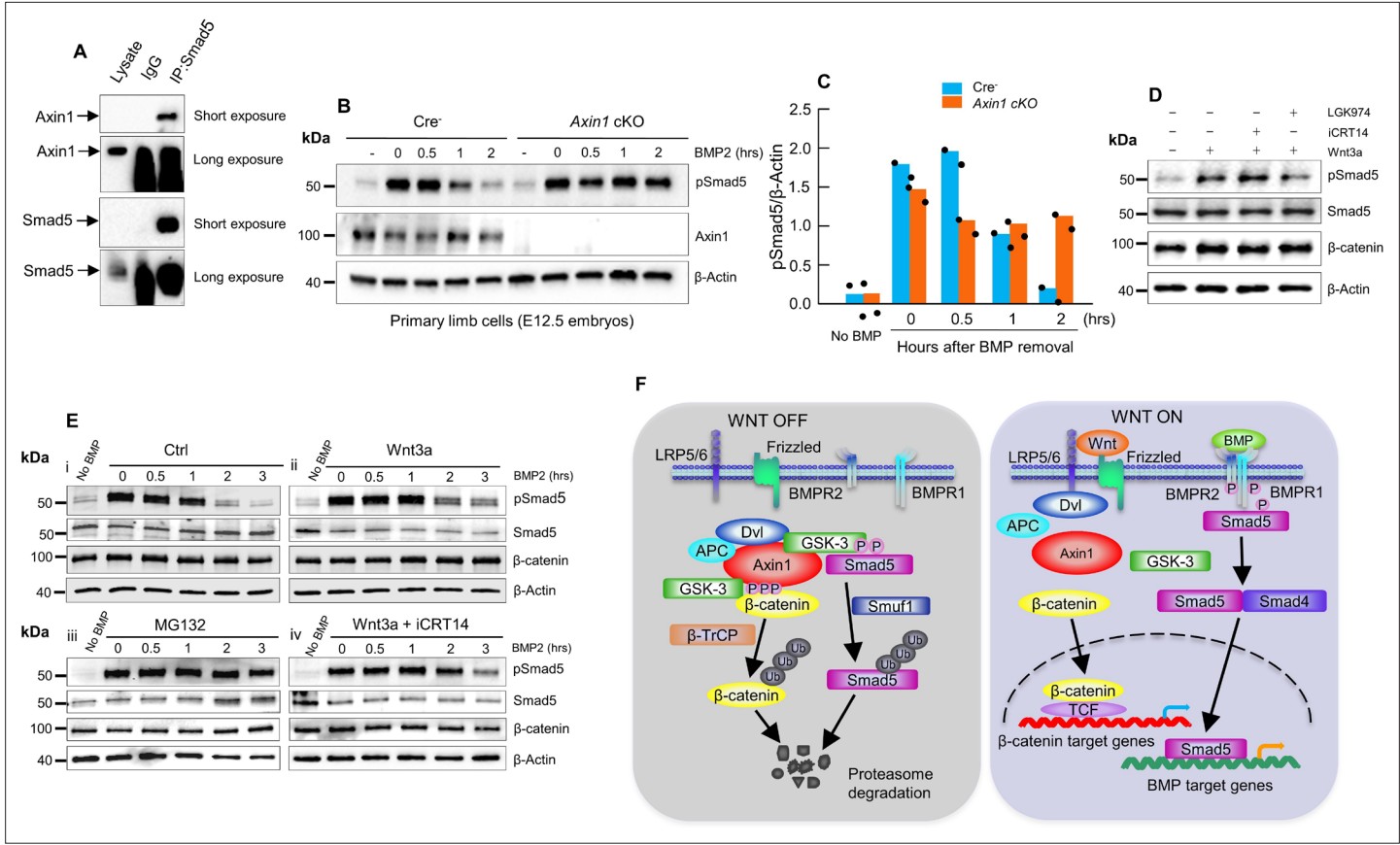

**Figure 6.** Axin1 regulates bone morphogenetic protein (BMP) signaling through increasing the degradation of pSmad5. (**A**) Interaction of endogenous Axin1 with Smad5 in C3H10T1/2 cells. Co-immunoprecipitation (co-IP) assay was performed using the anti-Smad5 antibody followed by Western blot analysis using the anti-Axin1 antibody (*Figure 6—source data 1*). (**B**) Pulse-chase experiments were performed showing that the duration of endogenous pSmad5 were extended in *Axin1*-deficient limb cells. Primary limb cells from Cre⁻ and *Axin1* cKO embryos (E12.5) were treated with BMP2 for 0.5 hr, followed by BMP2 wash-out. Cell lysates were extracted at different time points and analyzed by Western blot analysis using indicated antibodies (*Figure 6—source data 2*). (**C**) Quantification of the pSmad5 band (*Figure 6—source data 3*). (**D**) Wnt3a regulated Smad5 phosphorylation is independent of β-catenin. C3H10T1/2 cells were treated with Wnt3a for 1 hr in the absence or presence of iCRT14 or LGK974 and then harvested for Western blot analysis using indicated antibodies (*Figure 6—source data 4*). (**E**) Wnt3a prolonged the duration of pSmad5 in a β-catenin-independent manner. C3H10T1/2 cells were stimulated by BMP2 for 0.5 hr in the absence (i) or presence (ii) of the Wnt3a, proteasome inhibitor MG132 (iii), or the combination of both Wnt3a and iCRT14 (iv). Cells were harvested at the indicated time points after BMP2 wash-out and total cell extracts were analyzed by Western blot analysis (*Figure 6—source data 5–6*). (**F**) Model of integration of Wnt and BMP signaling pathways by Axin1. In the absence of Wnt stimulation, β-catenin is degraded by the destruction complex. Smad5 is also degraded by the Axin1 destruction complex (left panel). In the presence of Wnt ligands or in the absence of Axin1, β-catenin and pSmad5 degradation are inhibited, resulting in activation of both β-catenin and BMP/pSmad5 signaling (right panel).

The online version of this article includes the following source data for figure 6:

**Source data 1.** Original Co-IP results showed the interaction of endogenous Axin1 with Smad5 in C3H10T1/2 cells.

**Source data 2.** Original western blot of pSmad5 expression.

**Source data 3.** Quantification of pSmad5 levels in WT and *Axin1* conditional knockout (cKO) mice.

**Source data 4.** Original Western blot of pSmad5 and β-catenin expressions in C3H10T1/2 cells after treated with Wnt3a in the absence or presence of iCRT14 or LGK974.

**Source data 5.** Original Western blot of pSmad5, Smad5 and β-catenin in C3H10T1/2 cells treated by BMP2 in the absence (i) or presence (ii) of the Wnt3a.

**Source data 6.** Original Western blot of pSmad5, Smad5 and β-catenin in C3H10T1/2 cells treated by BMP2 in the presence of proteasome inhibitor MG132 (iii) and Wnt3a+ iCRT14 (iv).

*2013*). However, how Wnt ligands increase the stability of pSmad5 though Axin1 complex remains unknown. It has been reported that Wnt inhibits GSK-3β activity and prolongs the duration of BMP/pSmad1, leading to increased stability of Smad1 (*Fuentealba et al., 2007*).

It is worth mentioning that BMP signaling is downstream of β-catenin signaling during lower limb development, although we did find that Axin1 could directly interact with Smad5. This is because β-catenin inhibition could efficiently rescue FH phenotype in *Axin1* cKO mice and the presence of BMP signaling could not compensate the inhibition of β-catenin signaling, if we assume that BMP signaling is independent from β-catenin signaling. If Axin1 could regulate lower limb development through interacting with BMP signaling molecules completely independent from β-catenin signaling, inhibition of β-catenin signaling should not rescue the FH phenotype observed in *Axin1* cKO mice.

It has been suggested that FH occurrence is caused by genetic and environmental factors. At early pregnancy, the patients are exposed to harmful environments, such as radiation, toxic chemicals, virus infections, and specific drugs, which further leads to the transient upregulation of β-catenin-BMP signaling in limb mesenchymal cells. We still do not know which environmental factor(s) contribute(s) to the occurrence of FH disease. As a genetic factor, the epigenetic regulation of β-catenin-BMP signaling molecules in FH pathogenesis has not been explored. To fully understand the FH disease, further investigations need to be conducted.

In summary, in the present studies, we demonstrated that Axin1 is a key regulator of lower limb development. Therefore, we hypothesize that transient inhibition of Axin1 signaling or transient activation of β-catenin or BMP signaling during early skeletal development may be the cause of the FH disease. Our study could have significant impacts on the diagnosis and treatment of the FH disease.

# Materials and methods

## Mice

Conditional Axin1 loss-of-function mutant mice were generated by intercrossing double heterozygous for a floxed *Axin1* allele and the *Prrx1-Cre* transgenic allele (*Axin1^flox/wt^;Prrx1-Cre^+/-^*) (*Logan et al., 2002*) with homozygous floxed *Axin1* (*Axin^flox/flox^*) mice. Generation and characterization of *Axin1^flox/flox^* mice was previously reported (*Brault et al., 2001*). Both males and females of *Axin1^flox/flox^* mice are viable and fertile, and did not present any recognizable phenotype. Mice with floxed *Ctnnb1* (*β-catenin^flox/flox^*), in which exons 2–6 of the *Ctnnb1* gene are located within *loxP* sites (*Brault et al., 2001*), were obtained from Jackson Laboratory. All the mice were maintained under standard laboratory conditions with a 12 hr light/dark cycle. All animal procedures were approved by the Institutional Animal Care and Use Committee of Rush University Medical Center and all experimental methods and procedures were carried out in accordance with the approved guidelines.

For treatment with Wnt,β-catenin or BMP inhibitor, *Axin^flox/flox^* mice were crossed with *Axin1* cKO mice. The cross rendered 50% Cre-positive mice (*Axin1* cKO mice) and 50% Cre-negtive control mice (*Axin^flox/flox^*). The pregnant mice were injected with single dose of β-catenin inhibitor iCRT14 (2.5 mg/kg body weight, i.p. injection) or LGK974 (1.0 mg/kg body weight, i.p. injection) or BMP inhibitor dorsomorphin (2.5 mg/kg body weight, i.p. injection) at E9.5, respectively. The mice were sacrificed at E18.5 or 6 weeks of age. Dorsomorphin and iCRT14 were purchased from Tocris and LGK974 was purchased from Selleck.

## Radiographic and µCT analyses

Radiographs of mouse skeleton were taken after sacrifice of the animal using a Faxitron Cabinet X-ray system (Faxitron X-ray, Wheeling, IL, USA). For µCT analysis, bones were fixed in 10% buffered formalin, stored in 70% ethanol, and scanned using a Scanco VivaCT 40 system cone-beam scanner (Scanco Medical, Bassersdorf, Switzerland).

## Skeleton preparation and histology

Skeleton preparation and Alizarin red/Alcian blue staining were performed as described (*McLeod, 1980*). Briefly, mice were sacrificed using carbon dioxide, skinned, eviscerated, and fixed in 95% ethanol. Samples were placed in acetone to remove residual fat. Then the skeletons were stained by Alizarin red/Alcian blue. The stained skeletons were sequentially cleared in 1% potassium hydroxide. The cleared skeletons were transferred into 100% glycerol. For histology, samples were fixed in 10%

formalin, decalcified, and embedded in paraffin. Three µm sections were collected and stained with Alcian blue/hematoxylin and eosin (H&E) and Safranin O/fast green following standard procedure.

## Cell culture

For the primary limb cells, forelimb and hindlimb buds of 12.5 dpc *Prrx1-Cre*-negative or Cre-positive homozygous *Axin1*-flox embryos were collected in Hanks Balanced Salt Solution (HBSS, Sigma) and digested with 0.1% trysin, 0.1% collagenase in HBSS at 37°C for 15 min. The cells were disassociated through vigorous pipetting, spun down, resuspended in DMEM-F2, 10% FCS, and plated in six-well plates at $2 \times 10^7$ cells per well. Medium was changed daily.

C3H10T1/2 cells (CCL-226TM, ATCC, Manassas, VA, USA) were isolated from a line of C3H mouse embryo cells, and according to the manufacturer's instructions, the cell line was deposited by C Heidelberger and frozen on December 20, 2016. The identity has been authenticated including Ampule passage number, total cells/ampule (cell count using Trypan Blue staining method), growth properties (visual observation method), morphology (visual observation method), species determination, COI assay (interspecies), etc. The cell lines tested negative for mycoplasma contamination via the Hoechst DNA staining (indirect) method, Agar culture (direct) method-PCR-based assay. C3H10T1/2 cells were cultured in DMEM supplemented with 10% fetal bovine serum and 1% v/v penicillin/streptomycin. Cells were grown at 37°C and 5% $CO_2$ in a humidified incubator.

Prior to treatment with BMP2 (25 ng/ml, R&D systems), Wnt3a (100 ng/ml, R&D systems), cells were incubated in serum-free medium for 20 hr. Chemical inhibitors iCRT14 (25 M, Tocris) and MG132 (10 M, Sigma) were added 1 hr prior to BMP2 pulse. LGK974 (20 M, Selleck) was added 6 hr prior to BMP2 or Wnt3a addition.

## Western blot analysis

Cells were lysed with RIPA lysis buffer and protease inhibitor cocktail (Sigma #P8340). Protein concentration was determined by Pierce protein assay reagent (Thermo Fisher Scientific #1861426). Protein lysates were boiled in sample buffer (Bio-Rad #1610737). Protein samples were resolved on 10% precast gels (Bio-Rad #4568036) and transferred onto nitrocellulose membranes (Bio-Rad #1704159). Membranes were blocked in 5% blocking buffer and followed by incubation with primary antibodies and then detected with secondary antibody. The primary antibodies were specific for Smad5 (sc-7443), pSmad1/5 (CST #13820), β-catenin (sc-7963), Axin1 (Millipore #05-1579), Msx2 (sc-15396). Secondary antibodies were either HRP-conjugated goat anti-rabbit IgG (Bio-Rad #1706515) or rabbit anti-mouse IgG (Bio-Rad #1706516) and were revealed with Clarity western ECL substrate (Bio-Rad #1705061). Blots were exposed and scanned by ChemiDoc xRS + system (Bio-Rad).

**Table 1.** Names of genes and the primer sequences.

| Names of Genes | Primer Sequences |
| --- | --- |
| Axin1 Exon2-Forward | GAGCTCAGGGTCTGGAACAG |
| Axin1 Exon2-Reverse | CTGAGCTCTCTGCCTTCGTT |
| Bmp2-Forward | TGGAAGTGGCCCATTTAGAG |
| Bmp2-Reverse | TGACGCTTTTCTCGTTTGTG |
| Bmp4-Forward | TGAGCCTTTCCAGCAAGTTT |
| Bmp4-Reverse | CTTCCCGGTCTCAGGTATCA |
| Bmp7-Forward | GAAAACAGCAGCAGTGACCA |
| Bmp7-Reverse | GGTGGCGTTCATGTAGGAGT |
| Gremlin1-Forward | TGGAGAGGAGGTGCTTGAGT |
| Gremlin1-Reverse | AACTTCTTGGGCTTGCAG |
| Msx2-Forward | AACACAAGACCAACCGGAAG |
| Msx2-Reverse | GCAGCCATTTTCAGCTTTTC |

## Co-immunoprecipitation

The C3H10T1/2 cells were washed and collected with cold PBS, lysed in cold lysis buffer containing 50 mM Tris (pH 7.4), 150 mM NaCl, 1 mM EDTA, 1% Nonidet P-40, 10% glycerol, 0.5 mM DTT, protease inhibitor cocktail tablets (EDTA-free) (Roche) and phosphatase inhibitor cocktail tablet (Roche). The cell lysates were precleared with IgG-agarose beads (Sigma) for 8 hr at 4°C.

IP was carried out by incubating the cell lysates with anti-Smad5 antibody (Cell Signaling), rabbit IgG immobilized on Protein G Plus-Agarose bead (Santa Cruz) at 4°C overnight. The immunocomplexes were pelleted and washed with cold lysis buffer six times. The proteins were released from beads by boiling in SDS sample buffer, and the samples were analyzed by western blotting.

## Real-time PCR

Total RNA was isolated from cells or tissues by using RNeasy Mini Kit (QIAGEN). Reverse transcription was performed with iScript Reverse Transcription Supermix Kit (Bio-Rad). Real-time PCR was performed by a Bio-Rad SYBR Green kit and iCycler. Primers are listed in *Table 1*.

## Statistical analysis

Statistical analyses were performed using GraphPad Prism. Comparisons between two groups were performed using two-tailed, unpaired Student's *t*-test and one-way or two-way ANOVA followed by Tukey's post hoc test was performed to compare multiple groups under different genotypes or under different treatments at multiple time points. All data are presented as the mean ± SD. p values of less than 0.05 were considered statistically significant.

## Acknowledgements

We would like to express our gratitude to Ms Lily Yu for her help on processing and staining histological samples. This work was supported by the National Key Research and Development Program of China (2021YFB3800800) to LT and DC. This project was also supported by the National Natural Science Foundation of China (NSFC) grants 82030067 and 82161160342 to DC and grant 82172397 to LT and grant 81974320 to ZZ. DY was also supported by GuangDong Basic and Applied Basic Research Foundation grant 2021A1515111075.

## Additional information

### Competing interests

Di Chen: Reviewing editor, eLife. The other authors declare that no competing interests exist.

### Funding

| Funder | Grant reference number | Author |
| --- | --- | --- |
| National Natural Science Foundation of China | 82030067 | Di Chen |
| National Natural Science Foundation of China | 82161160342 | Di Chen |
| National Natural Science Foundation of China | 82172397 | Liping Tong |
| National Natural Science Foundation of China | 81974320 | Zeng Zhang |
| National Key Research and Development Program of China | 2021YFB3800800 | Liping Tong Di Chen |
| Guangdong Basic and Applied Basic Research Foundation | 2021A1515111075 | Dan Yi |

| Funder | Grant reference number | Author |
|---|---|---|

The funders had no role in study design, data collection and interpretation, or the decision to submit the work for publication.

## Author contributions

Rong Xie, Data curation, Investigation, Methodology, Writing - original draft; Dan Yi, Data curation, Funding acquisition, Methodology; Daofu Zeng, Data curation, Formal analysis, Funding acquisition, Investigation, Visualization, Methodology; Qiang Jie, Data curation, Formal analysis, Investigation, Writing – review and editing; Qinglin Kang, Zhenlin Zhang, Guozhi Xiao, Formal analysis, Investigation, Writing – review and editing; Zeng Zhang, Data curation, Supervision; Lin Chen, Formal analysis, Validation, Investigation, Writing – review and editing; Liping Tong, Data curation, Formal analysis, Funding acquisition, Validation, Investigation, Writing – review and editing; Di Chen, Conceptualization, Data curation, Formal analysis, Supervision, Funding acquisition, Validation, Investigation, Project administration, Writing – review and editing

## Author ORCIDs

Guozhi Xiao http://orcid.org/0000-0002-4269-2450
Di Chen http://orcid.org/0000-0002-4258-3457

## Ethics

This study was performed in strict accordance with the recommendations in the Guide for the Care and Use of Laboratory Animals of the National Institutes of Health. All of the animals were handled according to approved institutional animal care and use committee (IACUC) protocols (SIAT-IACUC-200302-YYS-CD-A1063) of the Shenzhen Institute of Advanced Technology, Chinese Academy of Sciences.

## Decision letter and Author response

Decision letter https://doi.org/10.7554/eLife.80013.sa1
Author response https://doi.org/10.7554/eLife.80013.sa2

# Additional files

## Supplementary files

• MDAR checklist

## Data availability

All data generated or analysed during this study are included in the manuscript and supporting file; Source Data files have been provided for Figures.

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
