## [Editor Report]

This manuscript uncovers the mesenchymal cells expressed Axin1 as a key regulator for Wnt and BMP signaling pathway which is essential for lower limb development. The data clearly demonstrates that inhibition of β-catenin and BMP signaling genetically and pharmacologically could largely reverse fibular hemimelia phenotype in mice. In general, the manuscript is clear, well-written, and concise and the study is well-structured and various techniques have been used to validate the data. It presents as a thorough study highlighting the importance of Axin1/ β-catenin/BMP signaling in FH development.

---

## [Decision Letter]

**Decision letter after peer review:**

Thank you for submitting your article "Specific Deletion of Axin1 Leads to Activation of β-Catenin/BMP Signaling Resulting in Fibular Hemimelia Phenotype in Mice" for consideration by *eLife*. Your article has been reviewed by 2 peer reviewers, and the evaluation has been overseen by a Reviewing Editor and Mone Zaidi as the Senior Editor. The reviewers have opted to remain anonymous.

Essential revisions:

Three issues were brought up in the review.

1. The deletion efficiency of Axin1 in mesenchymal cells should be determined.

2. Histology of bones (femur, tibia, knee joint) of Axin KO mice other than fibula should be provided.

3. All major comments under "Recommendations to Authors" need to be addressed.

*Reviewer #1 (Recommendations for the authors):*

This reviewer only has several points detailed below:

1. The deletion efficiency of Axin1 in mesenchymal cells should be determined.

2. Did the Axin1Prx1 KO mice carry the fibular deficiency phenotypes in both lower limbs?

3. Could deletion of both alleles of β-catenin in mesenchymal cells completely reduce the bone mass observed in Axin1Prx1 KO mice given deletion of one allele of β-catenin could only partially rescue this phenotype? And could iCRT14 treatment also reduce the bone mass in these mice other than rescue the fibular defects. These points can be addressed either experimentally or by discussion.

4. It is suggested that the joint fusion phenotype in Axin1Prx1 KO mice resemble to those found in noggin mutant mice, then could BMP inhibitor reverse this phenotype in Axin1Prx1 KO mice?

5. In Figure 5A, it appears that both Smad5 and Axin1 could be detected in the precipitated complex using negative control IgG.

*Reviewer #2 (Recommendations for the authors):*

1) Please include statistical analysis in the method section.

2) In human FH, the right fibula is more affected. Do authors see this in mice?

3) Please discuss why LGK974 had no effect while iCRT14 had an effect.

4) Beta-catenin and BMP inhibitor can rescue FH phenotype, separately. Do combined catenin and BMP inhibition have a better effect?

5) Fig 3. Do authors have data to show increased expression of beta-catenin signaling molecules or target genes in limb tissue from Axin1 cKO embryos as shown BMP signaling shown in Figure 4? Does Axin1 cKO have high bone mass phenotypes? Can this high bone phenotype be rescued by a beta-catenin inhibitor?

6) Fig 5. What Smad5 or pSmad5 protein levels in limb tissue from Axin1 cKO are? If an elevated BMP2 signal plays an important role in Axin1 null phenotype, Smad5 or pSmad5 protein levels should be increased because tissue should be exposed to endogenous BMP2.

7) Fig 5C, at 0.5 hr after BMP2 withdrew, pSmad5 levels rapidly decreased in Axin1 null cells compared to control, indicating that Axin1 is required for preventing pSmad5 degradation. At the later time points, pSmad5 levels is no change in Axin1 null cells but rapidly decreased in control cells, indicating that Axin1 is required for degradation. How authors explain this bi-phase effect?

---

## [Author Response]

Essential revisions:Three issues were brought up in the review.1. The deletion efficiency of Axin1 in mesenchymal cells should be determined.

We have determined the knockout (KO) efficiency in *Axin1* conditional KO mice as the reviewer suggested.

2. Histology of bones (femur, tibia, knee joint) of Axin KO mice other than fibula should be provided.

We have performed histology of lower limb, including femur, tibia and knee joint as the reviewer suggested.

3. All major comments under "Recommendations to Authors" need to be addressed.

We have addressed all major comments raised by 2 reviewers as the editor suggested.

Reviewer #1 (Recommendations for the authors):This reviewer only has several points detailed below:1. The deletion efficiency of Axin1 in mesenchymal cells should be determined.

We have determined the KO efficiency in *Axin1* conditional KO (cKO) mice as the reviewer suggested.

2. Did the Axin1Prx1 KO mice carry the fibular deficiency phenotypes in both lower limbs?

Yes, the fibular deficiency phenotype was observed in both side of the lower limbs in all *Axin1* cKO mice.

3. Could deletion of both alleles of β-catenin in mesenchymal cells completely reduce the bone mass observed in Axin1Prx1 KO mice given deletion of one allele of β-catenin could only partially rescue this phenotype? And could iCRT14 treatment also reduce the bone mass in these mice other than rescue the fibular defects. These points can be addressed either experimentally or by discussion.

Deletion of both alleles of *β-catenin* in mesenchymal cells (*β-catenin^flox/flox^;Prrx1-Cre*) causes embryonic lethal phenotype and mice died at the early embryonic stage. This is the main reason for us to switch genetic rescuing into using chemical inhibitor to analyze β-catenin signaling in FH phenotype.

4. It is suggested that the joint fusion phenotype in Axin1Prx1 KO mice resemble to those found in noggin mutant mice, then could BMP inhibitor reverse this phenotype in Axin1Prx1 KO mice?

We have tested the effect of BMP inhibitor and results demonstrated that BMP inhibitor indeed rescued FH phenotype observed in *Axin1* cKO mice.

5. In Figure 5A, it appears that both Smad5 and Axin1 could be detected in the precipitated complex using negative control IgG.

In this study, the IP was performed using the anti-Smad5 antibody followed by Western blot using the anti-Axin1 antibody in C3H10T_1/2_ cells. Smad5 and Axin1 was not detected in IgG control group so it is more likely that Smad5 and Axin1 were not precipitated by IgG.

Reviewer #2 (Recommendations for the authors):1) Please include statistical analysis in the method section.

We have included statistical analysis in the method section as the reviewer suggested.

2) In human FH, the right fibula is more affected. Do authors see this in mice?

In *Axin1* cKO mice, we see fibular defects in both sides of the limbs.

3) Please discuss why LGK974 had no effect while iCRT14 had an effect.

The LGK974 acts on Wnt secretion, the upstream of Axin1/β-catenin signaling. When *Axin1* is deleted, LGK974 could not affect β-catenin signaling.

4) Beta-catenin and BMP inhibitor can rescue FH phenotype, separately. Do combined catenin and BMP inhibition have a better effect?

We found that both β-catenin and BMP inhibitors can rescue FH phenotype and BMP seems acting downstream of β-catenin signaling. We hypothesize that combination treatment with both inhibitors may have similar effect as administration of β-catenin or BMP inhibitor alone. We will continue testing the combination effect of both inhibitors and compare with treatment with β-catenin inhibitor or BMP inhibitor alone

5) Fig 3. Do authors have data to show increased expression of beta-catenin signaling molecules or target genes in limb tissue from Axin1 cKO embryos as shown BMP signaling shown in Figure 4? Does Axin1 cKO have high bone mass phenotypes? Can this high bone phenotype be rescued by a beta-catenin inhibitor?

In the revised manuscript, we presented data of β-catenin upregulation in *Axin1* cKO mice. We did find high bone mass phenotype in *Axin1* cKO mice as shown in Fig. 2B, C. Deletion of *Axin1* causes the permanent change in bone mass, the treatment of a single dose of inhibitor could not rescue high bone mass phenotype.

6) Fig 5. What Smad5 or pSmad5 protein levels in limb tissue from Axin1 cKO are? If an elevated BMP2 signal plays an important role in Axin1 null phenotype, Smad5 or pSmad5 protein levels should be increased because tissue should be exposed to endogenous BMP2.

We agree with the reviewer. In *Axin1* cKO mice, both BMP2 and Smad5 levels were upregulated, leading to BMP signaling activation. As a result, inhibition of BMP signaling could efficiently rescue FH phenotype observed in *Axin1* cKO mice.

7) Fig 5C, at 0.5 hr after BMP2 withdrew, pSmad5 levels rapidly decreased in Axin1 null cells compared to control, indicating that Axin1 is required for preventing pSmad5 degradation. At the later time points, pSmad5 levels is no change in Axin1 null cells but rapidly decreased in control cells, indicating that Axin1 is required for degradation. How authors explain this bi-phase effect?

In Cre- negative cells, the degradation of pSmad5 is induced by Axin1/GSK-3β; in contrast, in *Axin1* cKO cells, pSmad5 degradation process is blocked so the steady-state pSmad5 levels are higher than those in Cre- negative cells.